# High prevalence of *Schistosoma mansoni* infection and stunting among school age children in communities along the Albert-Nile, Northern Uganda: A cross sectional study

Julius Mulindwa[1], Joyce Namulondo[2], Anna Kitibwa[2], Jacent Nassuuna[3], Oscar Asanya Nyangiri[2], Magambo Phillip Kimuda[2], Alex Boobo[2], Barbara Nerima[1], Fred Busingye[4], Rowel Candia[4], Annet Namukuta[4], Ronald Ssenyonga[5], Noah Ukumu[6], Paul Ajal[6], Moses Adriko[4], Harry Noyes[7], Claudia J. de Dood[8], Paul L. A. M. Corstjens[8], Govert J. van Dam[9], Alison M. Elliott[3], Enock Matovu[2]*, TrypanoGEN + Research group[¶]

1 Department of Biochemistry and Sports Sciences, College of Natural Sciences, Makerere University, Kampala, Uganda, 2 Department of Biotechnical and Diagnostic Sciences, College of Veterinary Medicine Animal Resources and Biosecurity, Makerere University, Kampala, Uganda, 3 Medical Research Council/ Uganda Virus Research Institute and London School of Hygiene & Tropical Medicine Uganda Research Unit, Entebbe, Uganda, 4 Vector Borne & NTD Control Division, Ministry of Health, Kampala, Uganda, 5 Department of Epidemiology and Biostatistics, School of Public Health, Makerere University, Kampala, Uganda, 6 Pakwach District Local Government, District Health Office, Pakwach, Uganda, 7 Centre for Genomic Research, University of Liverpool, Liverpool, United Kingdom, 8 Department of Cell and Chemical Biology, Leiden University Medical Center, Leiden, the Netherlands, 9 Department of Parasitology, Leiden University Medical Center, Leiden, the Netherlands

¶ Membership of the TrypanoGEN+ Research group of the H3Africa Consortium is provided in the Acknowledgments.
* matovue04@yahoo.com

**Editor:** JOANNE P. WEBSTER, Royal Veterinary College Department of Pathology and Infectious Diseases: The Royal Veterinary College Department of Pathobiology and Population Sciences, UNITED KINGDOM

## Abstract

### Background

Knowing the prevalence of schistosomiasis is key to informing programmes to control and eliminate the disease as a public health problem. It is also important to understand the impact of infection on child growth and development in order to allocate appropriate resources and effort to the control of the disease.

### Methods

We conducted a survey to estimate the prevalence of schistosomiasis among school aged children in villages along the Albert-Nile shore line in the district of Pakwach, North Western Uganda. A total of 914 children aged between 10–15 years were screened for *Schistosoma mansoni* using the POC-CCA and Kato Katz (KK) techniques. The infection intensities were assessed by POC-CCA and KK as well as CAA tests. The KK intensities were also correlated with POC-CCA and with CAA intensity. Anthropometric measurements were also taken and multivariate analysis was carried out to investigate their association with infection status.

**Data Availability Statement:** All relevant data are within the manuscript and its Supporting information files.

**Funding:** This study was funded by Wellcome Trust through the Science for Africa Foundation grant, ID H3A/18/004 awarded to EM and TrypanoGEN+ consortium. The funders had no role in the study design, data collection and analysis, decision to publish, or preparation of the manuscript.

**Competing interests:** The authors have declared that no competing interests exist.

## Results

The prevalence of schistosomiasis using the POC-CCA diagnostic test was estimated at 85% (95% CI: 83–87), being highest amongst children living closer to the Albert-Nile shore-line. Visual scoring of the POC-CCA results was more sensitive than the Kato Katz test and was positively correlated with the quantified infection intensities by the CAA test. The majority of the children were underweight (BMI<18.5), and most notably, boys had significantly lower height for age (stunting) than girls in the same age range (p < 0.0001), but this was not directly associated with *S. mansoni* infection.

## Conclusion

High prevalence of *S. mansoni* infection in the region calls for more frequent mass drug administration with praziquantel. We observed high levels of stunting which was not associated with schistosomiasis. There is a need for improved nutrition among the children in the area.

## Author summary

Schistosomiasis is a neglected but frequent disease that is caused by schistosomes, with over 290 million people worldwide at risk of infection. The major mode of transmission is through contact with fresh water sources infested with infected snails (the intermediate host). In this study, using the point of care test (POC-CCA), we screened 914 school aged children (10–15 years) living in the rural communities along the Lake Albert- River Nile shores of Pakwach district in Northern Uganda. We observed a very high prevalence of *S. mansoni* infections (over 80%) although the prevalence dropped to 40% in communities that were further from the lake shores. This high prevalence was also coupled with Kato Katz light schistosome infection intensities as categorised by WHO guidelines. We further compared the POC-CCA and Kato Katz tests to the more sensitive CAA assay and this revealed that even though both tests gave good probability of positive prediction, the POC-CCA had higher sensitivity in screening for *S. mansoni* infections than Kato Katz assay. The study also revealed high levels of stunting within the children, more so amongst boys. Frequent screening and mass treatment of these communities with praziquantel will reduce on the infection rates. But in addition, improved hygiene and sanitation will be required for a sustainable reduction in the prevalence and morbidity of schistosomiasis in the Albert-Nile communities along with dietary intervention for optimal child health.

## Introduction

Schistosomiasis (Bilharzia) is a neglected parasitic disease in humans caused by the blood flukes of the genus *Schistosoma*. It is widespread in tropical and subtropical regions with estimated transmission in over 78 countries and approximately 290 million people infected worldwide [1–3]. Approximately 280,000 annual deaths in Sub Saharan Africa have been attributed to schistosomiasis [4] and almost 1.9 million disability-adjusted life years [5].

In Uganda schistosomiasis is predominantly caused by *Schistosoma mansoni* with fewer cases of *Schistosoma haematobium* [6–10]. The former is transmitted when infected individuals defecate and release eggs into water bodies which then hatch and infect snails

(*Biomphalaria)*, that subsequently release cercariae which infect humans and the transmission cycle continues [11]. Most of the high infections in Uganda are amongst the shoreline communities of Lakes Albert, Victoria and Kyoga, and the Albert Nile [6,12]. The disease affects both children and adults, but the peak infection and intensity levels are in 10 to 20 year olds [13,14]. In order to achieve sustainable control and elimination of schistosomiasis, there is a need to improve the water, sanitation and hygiene (WASH) behavior so as to mitigate the risks of infection [15]. However, in Uganda the control programs have mainly focused on health education and preventive chemotherapy through mass drug administration (MDA) of praziquantel, in order to control morbidity, with minimal efforts to control environmental transmission [16]. MDA is often delivered to school-aged children (aged 5 to 15 years) and high-risk groups. The frequency of treatment depends on the focal prevalence of the disease according to World Health Organization (WHO) guidelines [17], and areas with low treatment coverage as observed previously [8]. In Uganda, some villages have had persistent high intensity and/or prevalence of schistosomiasis despite repeated MDA and these are referred to as persistent hotspots [18].

To determine the prevalence of schistosomiasis, the conventional standard for diagnosis of schistosomiasis as recommended by WHO is microscopic detection of *S. mansoni* eggs in faeces by the Kato Katz (KK) method and for *S. haematobium* using urine filtration [17,19]. The KK method is labour intensive but has low reagent cost and is widely used in low resource rural settings [20]. However the KK method has limited sensitivity especially in low endemicity areas where individuals with low or early schistosome infections often escape detection [21–24], resulting in underestimation of the disease prevalence. More sensitive techniques that detect actively secreted schistosome antigens in urine and serum have recently been developed. These include, the Point of Care-Circulating Cathodic Antigen (POC-CCA) test which specifically detects the *S. mansoni* CCA in urine [25,26] and the up-converting phosphor lateral flow (UCP-LF CAA) test which detects *Schistosoma* Circulating Anodic Antigen (CAA) in blood (serum, plasma) [27,28] and urine [29,30]. The POC-CCA is marketed as a qualitative rapid diagnostic test that has been extensively evaluated and recommended for surveillance and mapping of *S. mansoni* infections even in the low resource settings [31–36]. Misdiagnosis has been observed when calling trace results as positive [37] however this problem maybe more sever in low prevalence areas than in high prevalence ones [38]. Lastly, the UCP-LF CAA is an ultra-sensitive quantitative test with the potential to detect ultralow infections (down to a single worm pair) and early stage infection [39]. Measured CAA levels correlate with worm burden [27]. However, the UCP-LF CAA test cannot be used in the field due to a sample treatment steps that requires some basic laboratory equipment.

Despite annual MDA, the Lake Albert shoreline has often had high prevalence of infections among the primary school going children [40]. These children (aged 5 to 15 years) are the most exposed due to their responsibility for water-related household chores, and to behaviours such as swimming and bathing in water containing the infective cercariae [41]. These infections have been associated with adverse nutritional status among school-aged children [42–44] which might lead to cognitive impairment [45]. In Uganda, 30% of the children under five years are reported to be stunted, many of whom are in rural areas. Among the drivers of malnutrition in these areas has been the lack of access to clean water, poor sanitation, poor feeding practices and high disease burden [46]. However, a relationship between the burden of schistosomiasis infection and nutritional status and cognitive impairment has not been firmly established [47].

Therefore, the aim of this study was to determine the prevalence and intensity of *S. mansoni* infections and its association with the growth status of school going children (aged 10 to 15 years) living in recognised schistosomiasis hotspots at the shorelines of Albert Nile, Pakwach

district, Uganda. We compared the hotspots with non-hotspot sites and assessed the prevalence and intensity of *S. mansoni* infection between sexes, ages and locations. We further evaluated and compared the point of care (POC-CCA) diagnostic assay used for screening *S. mansoni* infections to the Kato Katz and CAA assays. This study was conducted as part of the TrypanoGEN+ consortium, one of whose overarching aims is to identify genetic markers for high schistosome burden in affected individuals (http://trypanogen.net).

## Methods

### Ethics statement

The study protocol was reviewed by the institutional review board of the Ministry of Health, Vector Control Division Research and Ethics Committee (Reference No. VCDREC106) and Uganda National Council of Science and Technology (Reference No. UNCST HS 118). The study was conducted with guidance from the district health officials, including the selection and training of the village health teams that were involved in the mobilisation and recruitment of the children into the study. The objectives, potential risks and benefits of the study were explained to the parents/ guardians who signed informed consent, and later explained to the school age children in English and Alur dialect who provided assent for participation into the study. Written formal consent from parents and written assent from the children were obtained. If a child was observed to have *S. mansoni* eggs in their stool, they were offered free treatment, which consisted of praziquantel at a dosage of 40mg/kg administered by trained Ministry of Health personnel, assisted by the district health worker.

### Study area and population

This was a cross sectional study conducted between October and November 2020, and the aim was to determine the prevalence of schistosomiasis among primary school going children living in recognised schistosomiasis high transmission hotspots in the West Nile sub-region district of Pakwach, Northern Uganda (Fig 1). Pakwach district is located along the western bank of the Albert Nile (Latitude:2.461944; Longitude:31.498333) as previously described [48–50]. It has an estimated population of 158,037 people (Uganda Bureau of Statistics UBOS, 2014) who are predominantly Alur speaking people of Nilotic ethnicity. The study was conducted in the sub counties of Pakwach, Panyimur, Panyango and Alwi. The study participants were school going children aged 10 to 15 years who were mobilized from the villages by the village health teams (VHTs) to a collection point, which was a school or health centre within the sub-county. The original plan was to collect at the schools from children attending in those schools, but this was not possible due to COVID19 related school closures. The primary schools that served as sites for screening included Panyigoro, Kivuje, Nyakagei, Kayonga, and Dei Health Center, which are located along Panyimur road and are in close proximity to the Albert Nile (1 km radius). Others included Pamitu and Alwi primary schools which are approximately 3 km and 10 km respectively from the Albert Nile (Fig 1) were included in the study to compare with the hotspot sites. The survey consisted of two main phases namely, the screening of the children for schistosomiasis using POC-CCA and the recruitment of eligible participants for sample collection.

### Screening for Schistosomiasis

The number of children to be surveyed per site was calculated using the Kish and Leslie formula for survey sampling in a cross sectional study [51]

$$n = Z^2(p)(1-p)/d^2$$

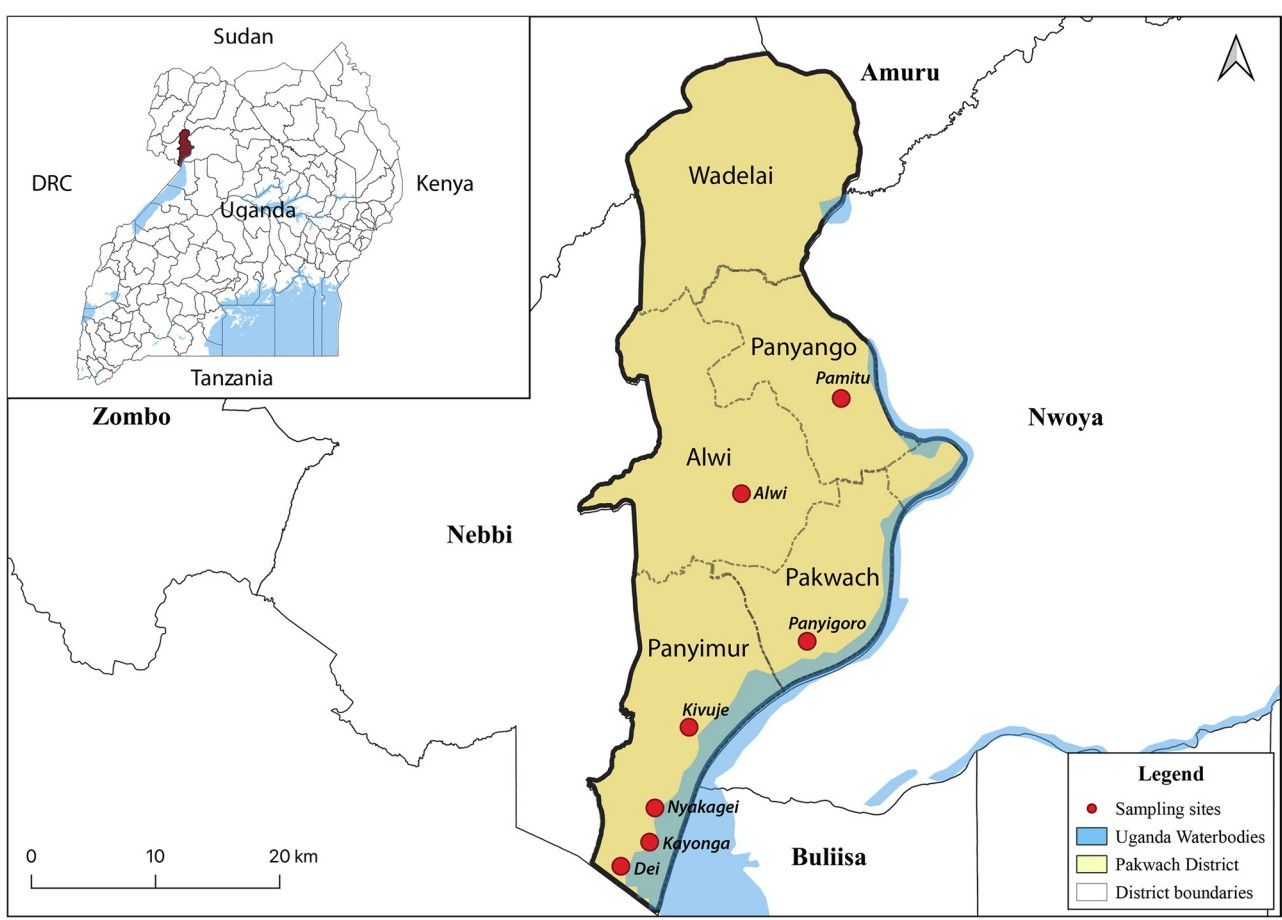

**Fig 1. Map showing the study sampling sites within the sub-counties of Pakwach district, West Nile, Uganda.** The base map was obtained from Uganda Bureau of Statistics (2012), http://purl.stanford.edu/vg894mz3698, and is in public domain with no restrictions on use.

where Z is the Z-score, p is the prevalence of disease and d is the precision or acceptable error in the estimate. Assuming Z of 1.96 (at 5% type 1 error), p of 0.5 (50% proportion with schistosomiasis in the study site), d of 10% and a sample size adjustment for non-response of 25%, we estimated the sample size required per site to be 128 children per site.

For each collection site, mobilization of children was done by the village health teams who moved around the village communities with information about the survey. Prior to the screening exercise, the children were sensitized and educated about schistosomiasis (Bilharzia), registered, and those who met the criteria (10 to15 years) were selected. For inclusion, each child was provided with a urine collection tube labelled with their specific registration code into which 10-20ml of fresh urine was collected. Those children that complied and brought the urine for testing were requested to return the following day with their parents for the recruitment exercise into the main survey. The urine was then immediately screened for *S. mansoni* infection CCA using the schistosomiasis POC-CCA rapid testing kit (Rapid Medical Diagnostics, Pretoria, South Africa, batch No. 191031120). Briefly, 2 drops (100μl) of urine were placed on the test cassette and incubated at room temperature for 20 minutes prior to visualisation. The intensity of the band in the test "T" area was scored using the G scores as described by Casacuberta-Partal et al. [52]; that is, 0 (G1), trace (G2, G3), 1+ (G4,G5), 2+ (G6,G7), 3+ (G8, G9) or 4+ (G10). We modified the G-score to include the 4+ to represent a measure of the

highest intensity (G10). An individual with a positive test for *S. mansoni* infection was referred to as a "case" and one with a negative test was referred to as a "control".

### Recruitment and sample collection

Before collection of study samples, the purpose was first explained to chairpersons of the villages and head teachers and pupils and political leaders and then later to the communities that took part in the study. Participants were selected for further study with the provision that informed consent from the parent/guardian and assent from the children, was first affirmed. From each participant, stool and blood samples were taken. For stool, the Kato Katz test was then carried out as previously described [19]. Briefly the Kato-Katz was done in duplicate and two technicians independently examined the duplicate slides and the average number of eggs per gram (EPG) calculated from the duplicate slide was determined as the infection intensity. The EPG was classified according to the WHO classification [17] as light infection (EPG < 100), moderate infection (EPG 100–399) and heavy infection (EPG $\geq$ 400).

Approximately 4 ml of venous blood were collected into an EDTA tube, centrifuged at 5000 rpm in order to separate the sample into packed cells and plasma. The plasma was aliquoted into a cryotubes and placed in liquid nitrogen for subsequent use in quantification of schistosome infection intensity by CAA test [27].

### Anthropometric measurements

Participants had their body weight, height and mid-upper arm circumference (MUAC) measurements taken. The body weight was measured in kilograms (kg) on a calibrated weighing scale, the height was measured in centimetres (cm) using a measuring tape from feet sole to the head top, and the MUAC was measured in cm using a standard MUAC tape. The height and weight measurements were converted into body mass index (BMI, kg/m$^2$). The Height for Age Z-score (HAZ) and the Body mass index for Age Z-score (BAZ) were determined using the WHO 2007 R package that incorporates the WHO child growth standards [53]. Stunting was defined as HAZ < -2 standard deviations (SDs) and wasting as BAZ < -2 SDs compared with the normal reference population.

### Measuring CAA levels in plasma

Schistosome infection intensity was further measured using the Circulating Anodic Antigen (CAA) levels in plasma as determined by the UCP-LF CAA test, that is, the SCAA20 assay format analysing the equivalent of 20 μl plasma [27,28]. Briefly 50 μl of plasma sample was extracted with 50 μl of 4% trichloroacetic acid (TCA): mixed by vortexing and incubated at room temperature for 5 min. The mixture was centrifuged at 13000 rpm for 5 minutes and 20 μl of clear supernatant incubated with 100 μl rehydrated CAA-UCP reporter conjugate in a 96 well plate for 1 hour at 37˚C with shaking at 900 rpm. CAA-specific lateral flow strips were then placed in each sample well to initiate flow (immunochromatography), left to dry and thereafter quantified using a Labrox Upcon scanner (Labrox Oy, Finland). Normalized signals Ratio (R) values were calculated by dividing test (T) signals by flow control (FC) signals, and expressed as CAA levels calculated in pg/ml against a CAA standard curve [54].

### Statistical analysis

The data collected on recruitment of the children comprised of, the disease detection and quantification (POC-CCA scores, Kato Katz and CAA), anthropometric measurements

(Height, Weight, Age and Sex) and the site of data collection were used for the quantitative analysis of *S. mansoni* prevalence and infection intensity, and nutritional status of the children. To achieve this, the data was computed using R 3.6.1 [55], Graph pad prism v9.0 software and STATA v15 Software.

In order to determine the overall and point prevalence, the epidemiological R packages epiR v2.0.19 exact method [56] and PoolTestR [57] were used respectively. For the point prevalence, adjustment for the hierarchical sampling structure was considered with sample maximum likelihood estimate of the prevalence with 95% confidence intervals and a Bayesian estimate with 95% credible intervals.

The intensity of *S. mansoni* infection as measured by the POC-CCA was compared between the gender, ages and sites using the Pearson Chi-square test. For all the statistical tests for significance or association the level of significance was $p < 0.05$. For *S. mansoni* infection intensity as quantified by the KK and CAA tests, the data distribution was first checked for normality and transformed accordingly (logarithmic) using the R gladder package. For both methods, the infection intensity by sex was compared using the Student's t-test and F-test whereas the comparison of infection intensity in the different ages and study sites was done using ANOVA. For all the statistical tests for significance or association the level of significance $p < 0.05$. The infection intensities as measured by POC-CCA, KK and CAA, were compared using the Spearman correlation test.

To determine the sensitivity, specificity and predictive values of the POC-CCA and KK tests for *S. mansoni*, the Stata_15.1 diagti tool was used. For this, the laboratory CAA test on the samples was used as the standard reference as it provides more sensitive confirmatory results [30]. To compare the anthropometric measurements (BMI, MUAC), age and sex between the boys and girls, Pearson correlation was carried out with two tailed P values considered (significant, alpha $<0.05$). In order to determine the z-scores for the anthropometric indicators for, height-for-age and BMI-for-age for the study children between 10–15 years, the R function who2007 was used [53]. The output z-scores for height-for-age (HAZ) and BMI-for-age (BAZ) were then flagged for analysing stunting, defined as HAZ $< -2$ standard deviations (SDs) and wasting as BAZ $< -2$ SDs. To determine the association between the nutrition status parameters (BMI, HAZ) and the infection intensity (CAA), linear regression analysis was done with age and sex considered as covariates. Given the hierarchical structure of the sample data, linear mixed regression analysis was carried out using the R package lmer [58] for which the model was fitted using the Restricted Maximum Likelihood criterion.

## Results

### Participant recruitment and sample collection

The survey was carried out at 7 sites located in 4 sub counties of Pakwach district (Fig 1). A total of 914 children aged 10–15 years were screened using the POC-CCA test. Of these, 727 (80%) assented to participate, following parent/guardian consent and were recruited into the study (Table 1 and Fig 2). The mean age of these recruited children was 11.99±1.67 years; 50% (364/727) were boys and the other half (363/727) were girls. The mean age for the girls was 11.89±1.64 and that for the boys was 12.09±1.69, although the age difference between them was not significant (P = 0.0948, t = 1.96). On comparing the number of children who consented to take part in the study verses those that were screened (727/914), we observed that Alwi and Pamitu had lowest percentage recruitment of 52% (62/118) and 47% (32/67) respectively.

**Table 1. Number of participants screened and recruited per study site.**

| Study site | POC-CCA screen | Infected cases | Uninfected controls | Recruited participants | Recruited cases | Recruited controls |
|---|---|---|---|---|---|---|
| Panyigoro | 162 | 143 | 19 | 156 | 141 | 15 |
| Kivuje | 134 | 132 | 2 | 102 | 100 | 2 |
| Nyakagei | 119 | 119 | 0 | 108 | 108 | 0 |
| Pamitu | 67 | 32 | 35 | 32 | 0 | 32 |
| Alwi | 118 | 55 | 63 | 62 | 0 | 62 |
| Dei | 175 | 169 | 6 | 144 | 140 | 4 |
| Kayonga | 139 | 133 | 6 | 123 | 117 | 6 |
| Total | 914 | 783 | 131 | 727 | 606 | 121 |

### Prevalence of Schistosomiasis

The POC-CCA assay has been considered more sensitive than KK for the screening of schistosomiasis [59]. We therefore determined the overall prevalence as the percentage of *S. mansoni* infected individuals among the total POC-CCA screened and point prevalence as percentage of those infected among individuals screened per site (Fig 3). The overall prevalence of schistosomiasis among the children tested from the different sites was 85% (783/914). The sites of Alwi and Pamitu had the lowest point prevalence (number of cases) of less than 47% at their respective sites whereas the other sites reported over 80% cases, with Nyakagei at 100% (Fig 3B). By adjusting for the hierarchical sampling structure in determination of the prevalence (S1 Table), we indeed observed that Nyakagei had the highest Bayesian estimated prevalence of 4% (95% CI: 0.003, 1.0) with Alwi having the lowest of 0.5% (95% CI: 0.004, 0.006). Alwi and Pamitu sites are located 14 km and 3 km respectively from the Albert Nile; whereas the sites of Panyigoro, Kayonga, Dei, Kivuje and Nyakagei are located within 1 km of the Lake/Albert Nile shores (S1A Fig). The prevalence of schistosomiasis at the study sites closer to the Nile was significantly different from those further from the Nile (p = 0.0084, t test = 12.7).

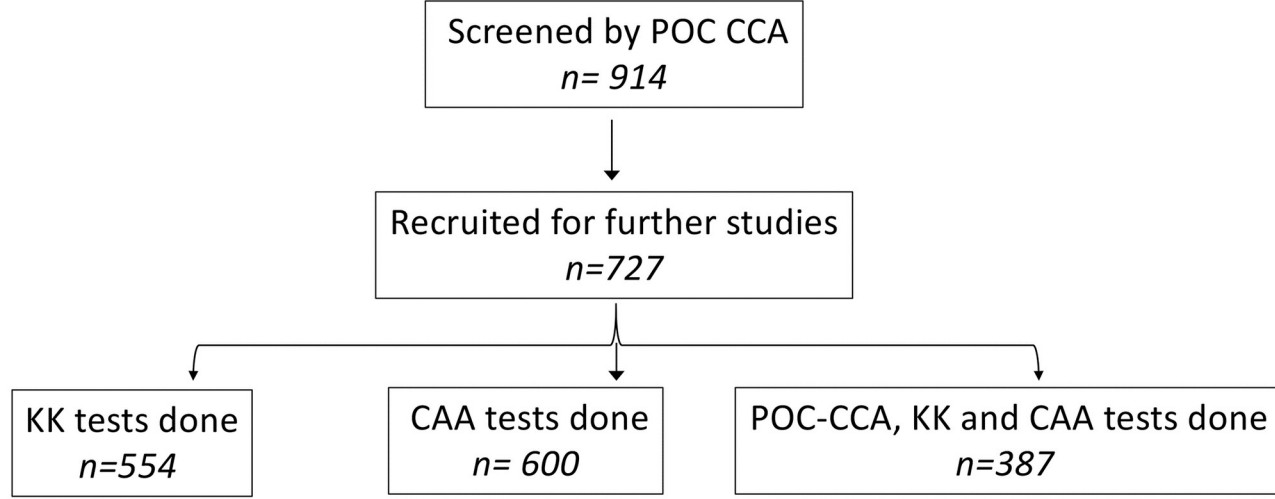

**Fig 2. Flow chart showing the number of participants screened, recruited into the survey and which *S. mansoni* detection tests were done on them.**

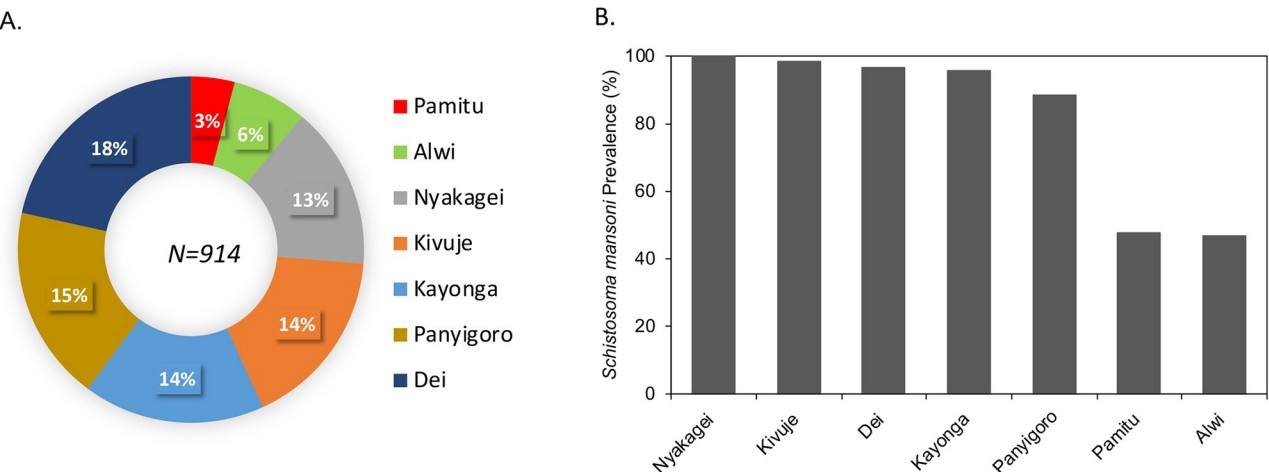

**Fig 3. Prevalence of *S. mansoni* among children aged 10–15 years in the study sites as detected by POC-CCA test. A**. The overall prevalence as a percentage of infected individuals among the total screened individuals (N = 914).**B**. The number of POC-CCA *S. mansoni* (Sm) positive cases in the individuals screened per site.

## Intensity of *S. mansoni* infections

From the 727 POC-CCA tested children that were recruited for the study, the positive band intensity from 571 children (78%) was visually scored on a scale of 0 to 4, whilst the remaining children the results were recorded just as positive or negative. Of 571 children with visual scores 80% (461/571) were considered positive (visual score >0.5[trace]) with a mean score of 2.53±1.0. There was no significant difference in the visual POC-CCA scores observed between the girls and boys (p = 0.413, $X^2$ = 3.94, df = 4); neither was there a significant association between age and POC-CCA intensity (Table 2). However, the mode POC-CCA visual score in all the age groups was observed to be 3+ (Fig 4A).

**Table 2. Summary descriptive statistics and between groups comparisons for the POC-CCA, Kato Katz and CAA test.**

|  |  | POC-CCA [0.5, 1, 2, 3, 4] | | Kato-Katz (EPG) | | | CAA (pg/ml x10e3) | | |
|---|---|---|---|---|---|---|---|---|---|
|  |  | N | ChiSq | N | Mean (SD) | Log[KK] | N | Mean (SD) | Log[CAA] |
| Sex | Females | 239 | P = 0.413 | 190 | 19(26) | P = 0.108 t = - -1.23 | 303 | 11(26) | P = 0.480 |
|  | Males | 222 | $X^2$ = 3.94 | 197 | 27(43) |  | 297 | 16(70) | t = -0.049 |
| Age | 10 | 125 | P = 0.361 | 97 | 18(20) | ANOVA P = 0.247 | 159 | 10(27) | ANOVA P = 0.047 |
|  | 11 | 72 | $X^2$ = 21.62 | 56 | 19(26) |  | 98 | 25(115) |  |
|  | 12 | 71 |  | 61 | 30(39) |  | 105 | 8(14) |  |
|  | 13 | 85 |  | 75 | 35(61) |  | 100 | 16(43) |  |
|  | 14 | 73 |  | 60 | 16(20) |  | 92 | 13(26) |  |
|  | 15 | 35 |  | 38 | 20(26) |  | 46 | 8(11) |  |
| Site | Dei | 138 | P = 1.01E-9 $X^2$ = 61.31 | 60 | 19(32) | ANOVA P = 0.940 | 137 | 33(106) | ANOVA P = 1.3E-27 |
|  | Kayonga | 117 |  | 58 | 20(22) |  | 115 | 9(24) |  |
|  | Kivuje | 98 |  | 76 | 23(33) |  | 96 | 6(7) |  |
|  | Nyakagei | 108 |  | 96 | 22(33) |  | 103 | 12(10) |  |
|  | Panyigoro | 0 |  | 97 | 28(49) |  | 133 | 5(10) |  |
|  | Alwi | 0 |  | 0 |  |  | 16 | 0.004(0.004) |  |

N = Number of Samples, SD = Standard Deviation, CL = Confidence level

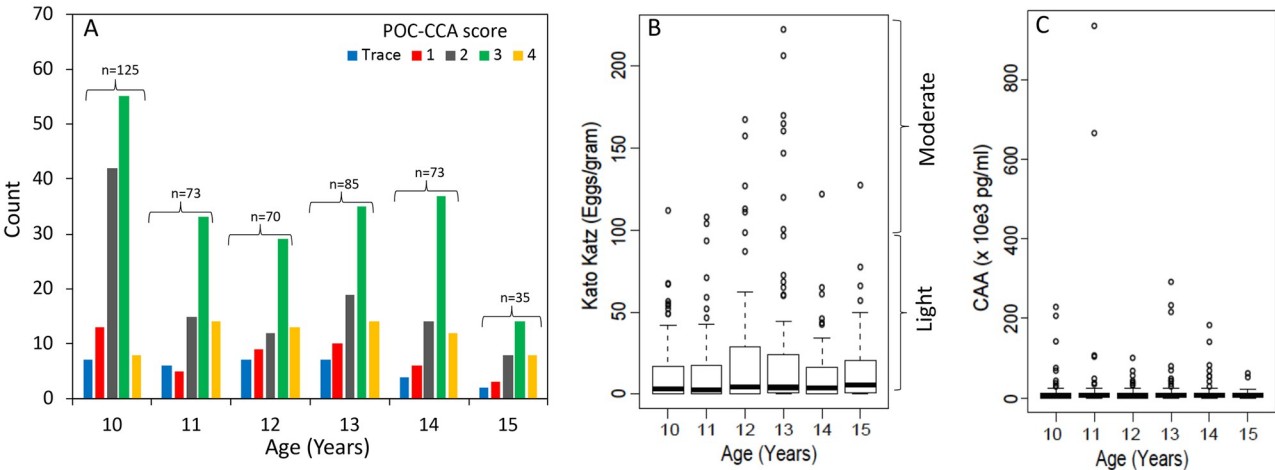

**Fig 4.** The distribution of infection intensity among the cases in the different age groups of the study participants as scored by **A**. POC-CCA (N = 461) scored as band intensities of trace, 1+, 2+, 3+ and 4+ of increased concentration of the circulating cathodic antigen of the worm; and **B**. Kato Katz (N = 387) measured by the mean number of eggs/gram of stool with infection intensity classified as light (EPG < 100), moderate (EPG 100–399) and heavy (EPG ≥ 400). **C**. CAA test done on plasma samples from 600 individuals whose urine had been screened in the field by POC-CCA.

The Kato-Katz analysis was carried out on 554 out of 727 recruited participants, of which 387 (70%) were positive as exhibited by presence of at least 1 *Schistosoma* egg on the two slides that were examined (mean EPG>0.5). From the 387 KK positive cases, we observed EPG ranging from 0.5–325 with a mean of 23±36.43 EPG. The majority of these KK positives were light infection (0–100 EPG) and only 5% (19/387) were moderate infections (Fig 3B). There was no observed difference in the mean EPG between the boys and girls (p = 0.108, t test = -1.23) and neither was there a difference in the KK infection intensities in the different age groups (ANOVA p = 0.247), nor the study sites (ANOVA p = 0.940). However the 13 year old children presented with the highest mean EPG.

The CAA analysis was carried out on 600 plasma samples that were collected from the POC-CCA positive cases (N = 577; including all KK positive samples) and negative individuals (N = 23). Of these, 64.5% (387/600) had been scored/tested by both POC-CCA and Kato Katz. Based on the SCAA20 cut-off thresholds in picogram per millilitre units (pg/ml) adapted from Corstjens et al [27], 84% (505/600) were positive (>30pg/ml), with CAA levels ranging from 30 to $9.34 \times 10^5$ pg/ml, 13% (77/600) were negative (< 15 pg/ml) and 3% (8/600) were indecisive (15-30pg/ml). However, for the analysis, we considered all samples with CAA<30pg/ml as negative.

We did not observe differences in the CAA infection intensities between the boys and girls (p = 0.48, t test = -0.049) (Table 2). However, there was a significant difference in the CAA intensities among the participant age groups (ANOVA p = 0.047), with the 11 year old group presenting with the highest CAA concentration in contrast to the EPG observed by Kato Katz (Fig 4C). These samples showing levels above 10,000pg/mL (the highest CAA standard included in the curve) may not be fully accurate as the SCAA20 format reaches a plateau. Furthermore, we observed significant differences in the CAA intensities in the study sites (Table 2, ANOVA p = $1.3 \times 10^{-27}$).

## POC-CCA precision

By comparing the children that had records for all three test parameters, that is, the semi-quantitative POC-CCA visual scores, the quantitative Kato Katz EPG and quantitative CAA

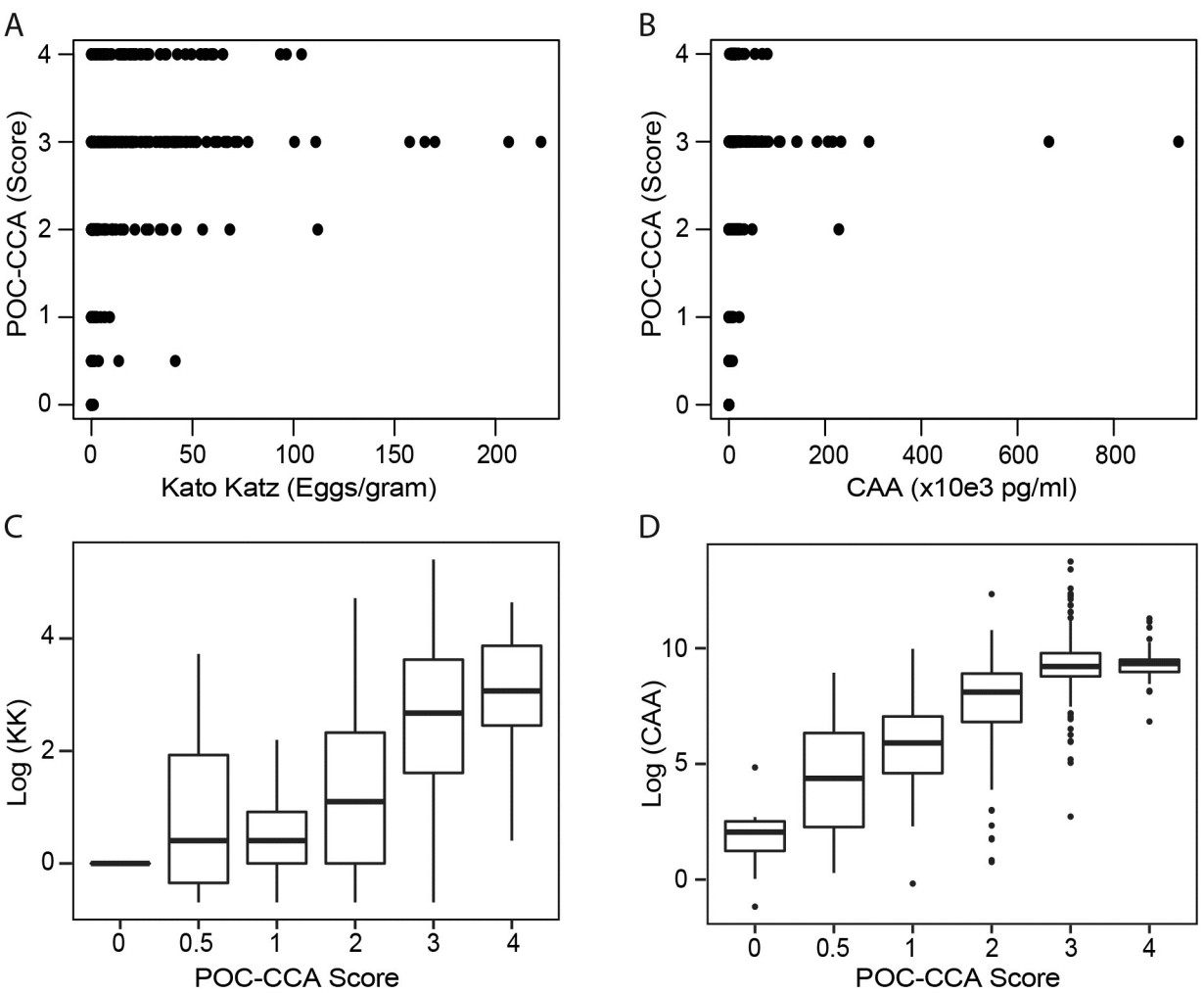

**Fig 5.** Relationship between POC-CCA and Kato Katz (**A, C**) and between POC-CCA and CAA (**B, D**). The original scale values for KK and CAA were used in panel **A, B** whereas the Log values were used in panel **C, D**.

concentrations (N = 387, S2 Fig), we observed positive correlation (r = 0.48–0.66) in the infection intensities between the methods (S2 Table and Fig 5). There was a higher correlation coefficient between POC-CCA and CAA (r = 0.66), than between POC-CCA and KK tests (r = 0.60); and a relatively low correlation between CAA and KK (r = 0.48). In order to determine the sensitivity of the field detection qualitative (POC-CCA) and qualitative (Kato Katz) tests, we compared the number of true/false positives and negatives with respect to the highly sensitive quantitative CAA test (S3 Table). The sensitivity of the POC-CCA test to correctly identify children with *S. mansoni* infection was 99% (95% CI: 98–99) whereas that for KK was 59% (95% CI: 54–64). However, the proportion of true negative correctly detected by the tests (Specificity) was higher for KK at 88% (95% CI: 75%– 95%) than for POC-CCA at 40% (95% CI: 27–56). When the POC-CCA trace bands were excluded, we observed an increased specificity to 77% (95% CI: 65–86). However, both tests showed high positive prediction of infection intensity with POC-CCA at 93% (95% CI: 90–95) and KK at 97% (95% CI: 94–99) (S3 Table and S3 Fig).

## Nutrition status and infection intensity

In order to assess the nutrition status of the study participants, we used the anthropometric scores of body mass index (BMI) and mid-upper arm circumference (MUAC). There was an overall high correlation between BMI and MUAC observed for all the participants recruited ($r = 0.65$, N = 715); the girls ($r = 0.71$, N = 356) and boys ($r = 0.69$, N = 359) (S4A and S4B Fig). However, there was a significant difference between the girls and boys with respect to both the BMI (F = 1.7, P = 8.12e-08) and MUAC (F = 1.5, 4.9e-05) (S4C and S4D Fig). The BMI for the boys ranged from 15.92 to 17.82 kg/m$^2$ whereas for the girls it ranged from 15.59 to 19.52 kg/m$^2$ (Table 3). The average BMI for the *S. mansoni* infected cases was 17.09±2.04 and that of the uninfected controls was 15.5±2.39. There was a significant difference in BMI between the cases and controls (F = 53.46, P = 7.103e-13, DF = 715) (S5A Fig) and this was also observed with the infection intensities (S5B Fig) and the sites of Pamitu and Alwi with predominantly uninfected individuals (S5C Fig). However, a multivariate analysis comparing BMI to infection intensity with age and gender as covariates showed a strong association between BMI and age (p <2e-16) and boys (p = 0.0495, S4 Table) but no association with infection intensity as measured by CAA. We observed no notable difference in the MUAC between the cases and control (S5D Fig).

We observed a significant difference in the height Z-score by age between the boys and girls but this was not seen for the BMI Z-score by age (S6B and S6D Fig). We further determined the Height for Age Z-score (HAZ) and BMI for Age Z-score (BAZ) as normalized values that are used as measures for stunting and wasting in children [60]. The mean HAZ and BAZ for the children studied were -1.91 ± 1.2 SD and -0.6 ± 1.01 SD, respectively. The overall prevalence of stunting (HAZ < -2SD) in the study sample (N = 714) was 47.9% (n = 342) whereas 7.7% (n = 54) were wasted (BAZ < -2 SD). More boys that were stunted 57% (204/358) than the girls 38% (138/356) (Fig 6A and Table 3); notably, the *S. mansoni* infected cases seemed more stunted than the control children (t-test p = 4.4e-10, Fig 6B).

**Table 3. Anthropometric measurements.**

| Age (Years) | N | BMI ± SD | MUAC ± SD | Stunting% (n) | 95% C.I | Wasting % (n) | 95% C.I |
|---|---|---|---|---|---|---|---|
| *Boys* | | | | | | | |
| 10 | 95 | 15.9 ± 1.6 | 18.6 ± 1.9 | 53.7 (51) | 43.1–64.2 | 6.3(5) | 0.9–11.7 |
| 11 | 54 | 16.1 ± 1.6 | 18.7 ± 1.7 | 42.6(23) | 28.5–56.7 | 9.3(3) | 0.6–17.9 |
| 12 | 55 | 16.6 ± 1.7 | 19.6 ± 1.4 | 47.3(26) | 33.2–61.4 | 9.1(3) | 0.6–17.6 |
| 13 | 65 | 17.0 ± 1.8 | 20.0 ± 1.7 | 67.7(44) | 55.6–79.8 | 9.2(3) | 1.4–17 |
| 14 | 52 | 17.8 ± 1.8 | 21.0 ± 1.8 | 73.1(38) | 60.1–86.1 | 7.7(3) | 0–15.9 |
| 15 | 37 | 17.6 ± 1.5 | 22.1 ± 1.5 | 59.5(22) | 42.3–76.6 | 8.1(2) | 0–18.3 |
| Total N | 358 | | | | | | |
| *Girls* | | | | | | | |
| 10 | 109 | 15.7 ± 1.5 | 18.5 ± 1.7 | 47.2(50) | 37.3–57.1 | 5.6(6) | 0.8–10.3 |
| 11 | 57 | 16.5 ± 1.7 | 19.7 ± 1.6 | 36.8(20) | 23.4–50.2 | 1.8(1) | 0–6 |
| 12 | 58 | 16.8 ± 2.5 | 20.5 ± 2.2 | 43.1(24) | 29.5–56.7 | 10.5(6) | 1.7–19.4 |
| 13 | 56 | 17.5 ± 2.5 | 21.7 ± 2.2 | 33.3(18) | 20.2–46.4 | 10.7(6) | 1.7–19.7 |
| 14 | 55 | 18.5 ± 2.5 | 22.5 ± 1.9 | 35.7(19) | 22.3–49.2 | 12.5(7) | 2.9–22.1 |
| 15 | 21 | 19.6 ± 1.7 | 23.9 ± 2.1 | 10(2) | 0–25.6 | 0 | 0–2.5 |
| Total N | 356 | | | | | | |

*Stunting is defined as HAZ < -2SD. Wasting is defined as BAZ < -2SD

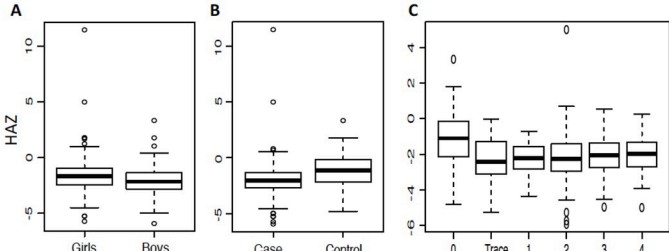

**Fig 6.** Height for Age Z score distribution by A. Sex, B. POC-CCA screen, C. POC-CCA infection intensity.

To further interrogate the possible association between infection status and the nutritional status, we did a multivariate linear regression analysis comparing the BMI (S4 Table), MUAC (S5 Table), Stunting, HAZ (S6 Table) and Wasting, BAZ (S7 Table) with age and gender as covariates. From this we observed a strong association between stunting (HAZ) and the male gender ($p = 9.86 \times 10^{-06}$) but with no association with infection intensity (CAA) nor age. In addition, there was no association observed between wasting (BAZ) and infection intensity (CAA) nor age or gender ($P > 0.05$). In order to test for the effects of infection intensity (as measured by POC-CCA, KK and CAA), other covariates and hierarchical structure on the stunting outcome, a linear mixed regression model was carried out (S8 Table). From this we observed that if one's gender was female, there would be a 0.16 unit reduction in the possibility of stunting; similarly, for a one unit increase in the age of the child would correspond to 0.37 reduction in stunting.

## Discussion

In Uganda, it was estimated in the year 2014 that 4 million people were infected with schistosomiasis, with 55% of the population at risk [61]. The latest (2019) national prevalence of intestinal schistosomiasis by POC-CCA in Uganda was reported as 25.6% [6]. In this study we focused on the endemic hotspots along the Albert-Nile in Pakwach district [6,14,50]. We screened (using POC-CCA) primary school going children (aged 10 to 15 years) at sampling sites in 4 sub counties, for which the estimated prevalence of *Schistosoma mansoni* in these hotspots was 85%. This prevalence was high and similar to that observed in the same region, that is 93% in children aged 10–14 years in Amor parish, Pakwach district [50] and 81.5% within the villages in the Rhino camp located along the Nile in Arua district [62]. This was higher than that observed in other studies along the L. Albert shore; in Piida fishing community, Butiaba during 1996–1997 at 72% prevalence [63]; similarly 67% prevalence was reported in Bugoigo fishing village, Buliisa district [40]. The high prevalence of schistosomiasis in these study sites was probably due to the inconsistent application of MDA, which had been last carried out in the year 2018 in this region (Communication from District Health Office, Pakwach). Another round of MDA was conducted in December 2020, after this study survey. In addition, the high infection intensities are often associated with the daily activities that bring people into contact with lake water contaminated with infective cercarie: that is, washing, bathing, fishing, sanitation and faecal disposal [41,64] and high disease virulence [65]. These regions with persistent transmission still pose a challenge to the control programs despite the MDA [18,66]. One of the challenges observed when carrying out this survey between October and November 2020 was the rising water levels of Lake Albert, which were reported to have displaced over 200 residents in Dei and Panyimur sub counties along the shoreline (S7 Fig).

Effects of climate change could have had an impact on schistosomiasis transmission dynamics and this would need further interrogation.

Field screening and determination of infection intensities plays an important role in the management of schistosomiasis morbidity and transmission [11]. A comparison between the qualitative POC-CCA and Kato Katz tests in the screening for *S. mansoni* in a rural field setting, showed the advantages and convenience of the less time consuming POC-CCA method over the Kato Katz. Indeed, on comparing these two field-screening tests with the more sensitive laboratory CAA test [29,30,67], the POC-CCA exhibited 1.5X higher sensitivity than the Kato Katz test even though both showed > 94% positive prediction of *S. mansoni* infection. A number of studies have shown the Kato-Katz test to have a low sensitivity in comparison to POC-CCA and this often underestimates the prevalence of infection [31,59,68–70]. Similar comparisons of the sensitivity of the POC-CCA to Kato Katz were also done in low endemic regions and revealed low sensitivity of the Kato Katz method [34,36,71]. Among school aged children living in low and high schistosomiasis transmission foci, low sensitivity of the Kato Katz test in comparison to the POC-CCA has also been observed [72] and this is probably due to the heterogeneous distribution of eggs within the stool sample that results in misdiagnosis of some infected individuals particularly those with low intensity infections [22,73]. Furthermore, given the positive correlation observed between the qualitative POC-CCA test and the more sensitive quantitative CAA test, field screening by visual scoring of the POC-CCA test seems a reliable method for the determining *S. mansoni* infection intensities in high endemic regions. However, there has been some concern that the much higher prevalence observed with POC-CCA than the KK might be due to a significant false positive rate with the POC-CCA. Interestingly, the POC-CCA was shown to have high sensitivity but low specificity when using the CAA as gold standard even though the specificity increased when samples with trace bands were excluded. This effect of trace bands giving false positive results has been reported [74] although it may be more of a problem in low endemicity settings [38]. An important limitation of the POC-CCA was its inability to detect *Schistosoma haematobium* [59] hence missing its prevalence in the population. And even though the CAA was a good gold standard in determining the sensitivity and specificity of the POC-CCA and KK methods, it is not practical in a field setting [30].

We further observed that the children living in villages within 1 km of the lake shore or river (Dei, Kayonga, Nyakagei, Kivuje, Panyigoro) had twice the prevalence of those living further away (>3km, Pamitu, Alwi) from the lake shore. This finding was in agreement with a previous study [63] showing that *S. mansoni* prevalence rates were higher in communities that were 5km or less from the shores of Lake Albert [13]. The close proximity to the lake and exposure to water contaminated with infective cercariae predisposes children to Schistosomiasis. This implies that, whereas MDA of praziquantel to school going children could prevent or reduce schistosomiasis morbidity and infection intensity, control measures focusing on sanitation, hygiene and snail control at the lake shores [75] could provide more lasting solutions to the transmission of the disease. The five river side communities had prevalences of 88–100% suggesting that the MDA programme conducted two years prior to our survey had little lasting effect on prevalence of schistosomiasis in this region. Studies of the effect of different praziquantel treatment frequencies in Lake Victoria islands showed the MDA reduced symptoms and the intensity of infection and prevalence as measured by KK, however prevalence measured by CCA remained constant [76]. Previous estimates of the impact of MDA on prevalence may have been biased by the low sensitivity of the KK test in low intensity infections, however any reduction in intensity of infection due to MDA can bring substantial health benefits despite not having much effect on prevalence.

BMI, is a marker for generalized adiposity and is the most widely used anthropometric measure as it is inexpensive and non-invasive and can be collected by evaluators such as the Village Health teams after receiving minimal training [77,78]. In addition, the MUAC can determine chronic energy deficiency as a measure of nutritional status [79,80]. A BMI less than 18.5kg/m$^2$ is considered as a sign of chronic energy and nutritional deficiencies [78]. In this study all the boys had a BMI <18.5 whereas the girls below 14 years also had low BMI implying that they were malnourished with chronic energy deficiency. The mean BMI was 0.77 standard deviations below the mean for World Health Organization (WHO) reference populations indicating significant levels of under nutrition in this area.

Furthermore, we observed high levels of stunting (47.9%) and moderate levels of wasting (7.7%) among the children. This phenomenon of under nutrition and stunting in schistosomiasis infected children has also been observed in other studies in Kenya [81] and Brazil [42]. This negative effect on the anthropometric status of school age children was also exhibited in the gender where the boys where more stunted than the girls. The mean height measures were within the lowest 1.2% of the WHO reference US and European distributions for each age group (mean height for age z-score– 2.24). This has also been shown in Brazil with gender differences in the growth of children with schistosomiasis [82]. But contrary to those studies, we did not find any association between schistosomiasis infection and stunting, even though we observed high levels of stunting (48%) in the study population. This could be attributed to the fact that approximately 33.9% of children under the age of 5 years in the West Nile region (includes Pakwach district) are stunted [83]. As a limitation, this study did not evaluate other factors in the environment that could be associated with stunting in these children, apart from testing for association with schistosomiasis. Furthermore, this cross-sectional study was conducted following the peak of the COVID19 pandemic during which period schools were locked down. If the schools were open, we probably would have recruited more children and probably not have missed those who were staying far from the collection sites or those that were engaged in other household activities during the survey.

**In conclusion**, this study showed the relevance of the urine POC-CCA assay in screening for schistosomiasis in rural settings and similar to previous findings, showed that it is more sensitive than the Kato Katz assay when the CAA is used as a reference standard. However, the POC-CCA is limited by its low specificity resulting mainly from the false positives from trace bands. From the selected high transmission hotspots along the shorelines in Pakwach district, we estimated a high prevalence of *S. mansoni* infections among school aged children. The impact of the high prevalence of schistosomiasis on child health is likely to be compounded by high levels of under nutrition and stunting especially among the boys. Therefore, we recommend regular screening with POC-CCA and consistency in the mass drug treatment with Praziquantel against *S. mansoni* infections. With regards to improving the nutrition status of the children, a nutritional assessment of dietary needs and possible supplementation should be undertaken with support from existing nutrition programs, hand in hand with awareness creation among the affected communities.

## Supporting information

**S1 Fig.  A**. The prevalence of *S. mansoni* cases per site compared to their relative distance from the Albert Nile. The prevalence was determined by the POC-CCA screening test using the overall sample size of 914. **B**. The infection intensity per site as determined by the CAA test. (TIF)

**S2 Fig.** Frequency distribution of the POC-CCA (A) visual scores, Kato Katz (B) and CAA (C) intensities in the study samples. Log transformed KK (D) and CAA (E) intensities.
(TIF)

**S3 Fig. Percentage of CAA positive tests among the POC-CCA visual scores (A) and Kato Katz (B) field tests.**
(TIF)

**S4 Fig. Anthropometric measurement of the body mass index (BMI) and mid-upper arm circumference (MUAC), comparisons between the Girls and Boys participating in the study.** The bars in plots C and D indicate the standard deviations from the mean.
(TIF)

**S5 Fig. Boxplots showing the distribution of BMI and MUAC with Condition (A,D), Infection intensity (B,E) and Study site (C,F).**
(TIF)

**S6 Fig. Distribution of Height and BMI with age among the study participants.** The age was compared with A. Height, B. Height Z-score, C. BMI and D. BMI Z-score. The vertical bars represent standard deviations from the mean.
(TIF)

**S7 Fig. Pictures of the Lake Albert shoreline highlighting the risen water levels in Pakwach district.** The author Julius Mulindwa was the photographer. **A**. Children and women taking water from the lake for household use. Picture was taken off the Dei sub county shoreline in November 2020. **B**. Submerged homes in Kayonga village, residents from this homestead were displaced.
(TIF)

**S1 Table. PoolTestR output statistics of prevalence estimates by Maximum likelihood and Bayesian model.**
(TIF)

**S2 Table. Spearman's rank correlation rho analysis between the test parameters.**
(TIF)

**S3 Table. Determination of the sensitivity and specificity of the field screening tests in comparison to the quantitative laboratory CAA test.**
(TIF)

**S4 Table. Linear regression model BMI vs CAA, Age, Sex.**
(TIF)

**S5 Table. Linear regression model MUAC vs CAA, Age, Sex.**
(TIF)

**S6 Table. Linear regression model HAZ vs CAA, Age and Sex covariate.**
(TIF)

**S7 Table. Linear regression model BAZ vs CAA, Age and Sex covariate.**
(TIF)

**S8 Table. Linear mixed model analysis on Stunting outcome.**
(TIF)

## Acknowledgments

We do acknowledge and thank all the children and the parents/guardians that participated in this study. We do appreciate the efforts by the Village health team members and the local council administrators of the villages of Panyigoro, Kivuje, Nyakagei, Kayonga, Dei, Pamitu and Alwi. We do acknowledge Dr. Pytsje T Hoekstra for the CAA analytical training and Dr. Miriam Casacuberta for the POC-CCA training.

Membership of the TrypanoGEN+ Research group of the H3Africa Consortium:

Annette MacLeod, Bruno Bucheton, Gustave Simo, Dieudonne N. Mumba, Mathurin Koffi, Ozlem T. Bishop, Pius V. Alibu, Janelisa Musaya, Christiane Hertz-Fowler.

## Author Contributions

**Conceptualization:** Julius Mulindwa, Alison M. Elliott, Enock Matovu.

**Data curation:** Julius Mulindwa, Joyce Namulondo.

**Formal analysis:** Julius Mulindwa, Ronald Ssenyonga, Harry Noyes.

**Funding acquisition:** Alison M. Elliott, Enock Matovu.

**Investigation:** Julius Mulindwa, Joyce Namulondo, Anna Kitibwa, Jacent Nassuuna, Oscar Asanya Nyangiri, Magambo Phillip Kimuda, Alex Boobo, Fred Busingye, Rowel Candia, Annet Namukuta, Noah Ukumu, Paul Ajal, Moses Adriko, Claudia J. de Dood, Paul L. A. M. Corstjens, Govert J. van Dam.

**Methodology:** Julius Mulindwa, Joyce Namulondo, Anna Kitibwa, Jacent Nassuuna, Claudia J. de Dood, Alison M. Elliott, Enock Matovu.

**Project administration:** Barbara Nerima.

**Supervision:** Julius Mulindwa, Moses Adriko.

**Writing – original draft:** Julius Mulindwa.

**Writing – review & editing:** Julius Mulindwa, Joyce Namulondo, Oscar Asanya Nyangiri, Magambo Phillip Kimuda, Moses Adriko, Harry Noyes, Paul L. A. M. Corstjens, Govert J. van Dam, Enock Matovu.

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
