## [Decision Letter · Decision Letter 0]

1 Sep 2021

Dear Dr. Matovu,

Thank you very much for submitting your manuscript "High levels of Schistosoma mansoni infection and stunting among school age children in communities along the Albert-Nile, Northern Uganda" for consideration at PLOS Neglected Tropical Diseases. As with all papers reviewed by the journal, your manuscript was reviewed by members of the editorial board and by two highly experienced independent reviewers. In light of the reviews (below this email), we would like to invite the resubmission of a significantly-revised version that takes into account all the reviewers' comments, including additional statistical analyses and re-formatting the manuscript where applicable.

We cannot make any decision about publication until we have seen the revised manuscript and your response to the reviewers' comments. Your revised manuscript is also likely to be sent to reviewers for further evaluation.

[1] A letter containing a detailed list of your responses to all the review comments and a description of the changes you have made in the manuscript. Please note while forming your response, if your article is accepted, you may have the opportunity to make the peer review history publicly available. The record will include editor decision letters (with reviews) and your responses to reviewer comments. If eligible, we will contact you to opt in or out.

Sincerely,

Joanne P. Webster

Associate Editor

Christine Budke

Deputy Editor

Reviewer's Responses to Questions

**Key Review Criteria Required for Acceptance?**

**Methods**

-Are the objectives of the study clearly articulated with a clear testable hypothesis stated?

-Is the study design appropriate to address the stated objectives?

-Is the population clearly described and appropriate for the hypothesis being tested?

-Is the sample size sufficient to ensure adequate power to address the hypothesis being tested?

-Were correct statistical analysis used to support conclusions?

-Are there concerns about ethical or regulatory requirements being met?

Reviewer #1: 1. Title: add that this is a cross-sectional study

2. Abstract: the number of children examined for KK was < 914. Mention that Intensity of infection was assessed by POC-CCA and KK as well as CAA. Mention that KK intensity was also correlated with CAA and with CCA intensity. 

3. Author summary: intensities of infections were light, according to WHO guidelines.

4. Introduction:

5. Line 106: strictly speaking a viable egg is one that goes on to develop into the next stage. KK enables to detects eggs, but it cannot distinguish between viable and non-viable eggs. I suggest that the word “viable” is removed.

6. Lines 116-118: pros and cons were mentioned for KK and CAA but not for POC-CCA. It might be worth mentioning current concerns associated with POC-CCA (e.g. Peralta and Cavalcanti 2018)

7. Lines 101 and 124 report different ages from school aged children (5-15) and (6-15)

8. Lines 125-128: the relationship between schistosomiasis infection and nutritional status and cognitive impairment has not been firmly stablished yet (e.g. Welch et al. 2017). I suggest that this is acknowledged

9. Line 130: as this is a cross-sectional study, causal relationships cannot be ascertained. I suggest that the word “affect” is replaced with “association”.

10. Line 155: are the four sub-counties visited representative of the whole district? E.g. How many sub-counties are there in Pakwach? If there are more than 4 sub-counties, how did they choose to survey these 4 sub-counties? 

11. Line 157: are the children representative of the sub-counties? E.g. What was the sampling frame for the collection points? How were the collection points selected? Is there a possibility that some children in the district could not reach the collection points and that the results could be biased to reflect the prevalence around the collection points rather than the prevalence of the whole district?

12. Line 164: it was not clear upon first read of the paper that the survey comprised two stages (a screening phase and the main survey phase). I suggest that this is clearly outlined before explaining the screening process. I suggest that the reasons why the children were asked to return the next day with the parents are detailed (i.e. to carry out main survey?). Would having to return the next day have impacted the representativeness of the sample? Is it possible that those children living far from school could not return to the collection site on the second day?

13. Line 168 indicates that POC-CAA was assessed for all 10–15-year-old children. Line 183 indicates that participants were randomly selected. Does this mean that all the children that turned up in day one were surveyed and only some of them were asked to return the next day? How was the required sample size estimated and how was the random selection conducted?

14. Line 185: was POC-CCA measured on day 1 while CAA, KK and anthropometric measurements were taken on day 2?

15. Lines 243-4 indicated that 80% of the POC-CCA screened children assented to participate. It is ambiguous whether consent and assent for POC-CCA had been granted or not. This needs to be specified. 

16. Line 209: do they know the precision of the weighing scale? How was aged obtained (how reliable is this variable)?

17. Lines 233-4: the way that overall prevalence (and district summary statistics) is calculated gives more weight to those locations that had more participants and does not take into account the hierarchical structure of the survey design. A way to overcome this could be to estimate prevalence using random effects for collection sites, e.g. with R package lme4. The R “survey” package can also be used.

18. Lines 235-238: t-tests, ANOVA and linear regression are valid when the underlying assumptions hold. Have the authors checked that this is the case? Spearman correlation was carried out but this method is not listed in this section. The authors use “95% confidence level (CL)” in Table 2, line 280, and other places. What is 95% CL? How should it be interpreted? They have not defined what level they consider as significant. E.g. is it 0.05 or 0.1? I suggest that p-values smaller than 0.001 are reported as “<0.001” instead of reporting all the decimal figures.

19. To avoid misinterpretations I suggest that the authors define what they mean by “phenotype”, “case” and “control”.

Reviewer #2: Overall the study design was appropriate and the study population has been clearly described.

However, addressing the balance of the article across the stated objectives would strengthen the work.

For example, the authors begin with 'Efforts to control schistosomiasis infection in Uganda have been greatly affected by the limited knowledge of the extent of infection, impact of disease on child growth and the inconsistency in delivery of treatment through mass drug administration.'

Nevertheless, the majority of the article compares the relative merits of screening tools for S. mansoni. While this is valuable, this aim needs to be better explicated in the summary and introduction and the implications better detailed in the discussion/conclusion. 

The authors found an 85% prevalence of S.mansoni among the children involved. Yet these children were apparently participating in an annual MDA programme. Further information about the MDA programme, the uptake among the children and indeed, the potential reasons for the failings of this public health intervention would be of interest to both general readers and policy makers.

**Results**

-Does the analysis presented match the analysis plan?

-Are the results clearly and completely presented?

-Are the figures (Tables, Images) of sufficient quality for clarity?

Reviewer #1: 20. Line 242: should it say 4 sub-counties?

21. Line 248: consider including a flow diagram with the number of children that were assessed for each diagnostic test: e.g. 914 children that provided POC-CCA samples, 727 children were recruited for further studies of which 554 were analysed for KK and 600 for CAA; 387 provided samples for all three tests. 

22. Lines 249-255 are methods rather than results and not all of them apply to this study. I suggest that the authors consider removing these lines. It might be worth mentioning the TrypanoGEN+ study in the introduction and contextualise this manuscript within the TrypanoGEN+ study. 

23. Lines 258 and 262: as already mentioned, I suggest that the authors consider using hierarchical modelling to estimate population prevalence. 

24. Line 260: do the authors mean 4 to 15 or 10 to 15?

25. Line 261: epiR was used to estimate the confidence interval for prevalence but the authors do not give details about the statistical method used in the calculation. For example, was it calculated using Wilson’s method or the exact method?

26. Intensity of infection: 

a) As intensity was measured in three different ways, and to avoid confusion, when mentioning intensity, it would be useful to refer to the method used (i.e. kk, POC-CCA or CAA)

b) Check that the statistical tests used (t-test and ANOVA) can be used i.e. that the assumptions hold. If they don’t hold then consider transforming the data or using non-parametric tests

c) POC-CCA scores might be better assessed with chi squared tests than by comparing means

d) When reporting ratios, also include their confidence intervals (lines 291 and 304)

e) The p-value in line 303 of 0.096 is considered to be significant. Stablish in the methods the p-value threshold that will be used throughout the paper

f) Consider applying hierarchical multivariable regression

Correlation:

27. Line 318: although the correlation coefficient of 0.48 is the lowest of the three reported, it is technically not “low”. 

Sensitivity/specificity:

28. Note that 6 KK positive cases were CAA negative. Comment on the implications of this with respect to the estimated sensitivity/specificity

29. Report confidence intervals for sensitivity and specificity estimates

30. Line 346: says in the study population. However, the reported value is a sample result.

31. Line 336 refers to POC_CCA intensity (Fig S5b). However, the multivariable analysis was done on CAA intensity. I suggest that theses tests are done on the same intensity results, either POC-CCA or CAA.

32. Line 340: for completeness, it might be good to do a multivariable test on MUAC.

Tables and figures:

37. Table 2 caption: “Summary descriptive statistics and between groups comparisons for the POC-CCA, Kato Katz and CAA test”.

38. Figure 1: would it be possible to add the sub-county names and border lines? 

39. Figure 2: “Schistosoma mansoni”

40. Figure 3A. Numbers add up to 83%. Shouldn’t this be 100%?

41. Figure 4: as no correlations are reported on the graphs, would it be more appropriate to label this graph as “relationship between…” instead of “correlation”? I would suggest changing “absolute measured values” to “original scale values”.

Supplementary data:

42. Figure 1: mention what test was used to determine prevalence and the overall sample size. A: Make “S. mansoni” on the y-axis italics. B consider using log-scale for the y-axis values

43. Figure S3. What would a similar plot for CAA and KK look like?

44. Figure S4. What do the vertical bars represent? Are they confidence intervals?

45. Figure S6. What do the vertical bars represent?

46. Tables S4 to S6: Is the first row in each table the intercept? Would it be possible for the tables to include the regression coefficients?

Reviewer #2: See above.

The authors findings regarding the varying prevalence of Schistosomiasis among children living in a radius from the Albert-Nile is important. Yet little detail is provided as to potential reasons for these differences or indeed, if particular sub-populations of children had greater/lower risk i.e. presumably children who have consistently attended school even in high risk areas will have had lower exposure rates? 

The analysis in relation to child stunting while interesting, lacks nuance. Identifying the potential interactions between the drivers of child stunting and Schistosome infection across this population would enhance this impact and usefulness of this work.

Equally, when exploring child stunting, the KK results were not presented. Again, following through on the current frame of the article around the comparison of the three screening tools, presenting a more complete analysis here would be of interest.

**Conclusions**

-Are the conclusions supported by the data presented?

-Are the limitations of analysis clearly described?

-Do the authors discuss how these data can be helpful to advance our understanding of the topic under study?

-Is public health relevance addressed?

Reviewer #1: The conclusions are supported by the data presented.

The limitations are not fully covered (see comments above).

The public health relevance is addressed.

Reviewer #2: The authors findings regarding the prevalence of Schistosomiasis among populations living a varying distance from the Albert-Nile is important. Many elements detailed in the discussion/conclusion with regard to public health measures, WASH elements, climate change etc. are not dealt with in the main body of the work.

**Editorial and Data Presentation Modifications?**

Reviewer #1: (No Response)

Reviewer #2: (No Response)

**Summary and General Comments**

Reviewer #1: This is a very complete study covering a range of diagnostic techniques for schistosomiasis that also includes a nutritional status assessment. My suggestions are oriented towards a better understanding the representativeness of the sample and the statistical methods used. I also made recommendations in relation to the literature coverage with respect to POC-CCA pros and cons and to the relationship between schistosomiasis infection and stunting. I would highlight the relevance of the following findings: after years of annual treatment prevalence in this area remains high, although intensity of infection is mainly light; in the sample examined the was no evidence of association between schistosomiasis and nutritional status; and the three measures of infection intensity evaluated were significantly correlated with each other. As the survey design is hierarchical, I encourage the authors to apply statistical hierarchical methods. As checking whether the assumptions of statistical test hold is a critical step in data analysis, I would recommend that the authors report whether these checks have been carried out.

Reviewer #2: This is a vitally important topic. The findings of the point-prevalence survey of Schistosomiasis among school-age children living along the Albert-Nile river basin in Uganda are useful. 

Nevertheless, there is a tension in this article between the comparative analysis of the three screening tools and the wider aims of the study. 

If the authors choose to focus on the aims as described, then critical elements of the context are missing: from the particular WASH factors involved to the wider prevalence of stunting among boys in Uganda to the reasons the current annual MDA is apparently failing (is it uptake, resistance or indeed frequency?). The exploration of the potential relationship between Schistosomiasis and child stunting again is a very valuable aim but the analysis is superficial and the discussion lacks reference to the wider context of child stunting in Uganda. Addressing these elements would strengthen the impact of the work on public health policy and practice in Uganda and wider.

PLOS authors have the option to publish the peer review history of their article (what does this mean?). If published, this will include your full peer review and any attached files.

Reviewer #1: No

Reviewer #2: No
---

## [Decision Letter · Decision Letter 1]

15 Mar 2022

Dear Dr. Matovu,

Thank you very much for submitting your revised manuscript "High levels of Schistosoma mansoni infection and stunting among school age children in communities along the Albert-Nile, Northern Uganda: a cross sectional study" for consideration at PLOS Neglected Tropical Diseases. As with all papers reviewed by the journal, your manuscript was reviewed by members of the editorial board and by independent reviewers. In light of the reviews (below this email), whilst we appreciate that improvements to the text have been made, we would like to invite the further resubmission of a significantly-revised version that takes into account the reviewers' comments. In particular, please ensure you carefully respond to and accommodate fully each of the highly valid and extremely useful further comments provided by referee 1 - which includes those highlighting where your responses to date (such as regarding the statistical analyses used here) have not sufficiently rectified the issues highlighted before.

We cannot make any decision about publication until we have seen the revised manuscript and your detailed response to the reviewers' comments. Your revised manuscript is also likely to be sent to reviewers for further evaluation. However, please also note that, in order to minimize the risk of reviewer fatigue, that this will be the final opportunity to resubmit here.

Sincerely,

JOANNE P. WEBSTER

Associate Editor

Christine Budke

Deputy Editor

Reviewer's Responses to Questions

**Key Review Criteria Required for Acceptance?**

**Methods**

-Are the objectives of the study clearly articulated with a clear testable hypothesis stated?

-Is the study design appropriate to address the stated objectives?

-Is the population clearly described and appropriate for the hypothesis being tested?

-Is the sample size sufficient to ensure adequate power to address the hypothesis being tested?

-Were correct statistical analysis used to support conclusions?

-Are there concerns about ethical or regulatory requirements being met?

Reviewer #1: -Are the objectives of the study clearly articulated with a clear testable hypothesis stated?

Some objectives are clearly articulated but (at least) the following are missing: comparison of hotspot sites with non-hotspot sites (lines 177-179), assessment of differences in prevalence and intensity between, sexes, ages and locations.

Introduction: not enough is said about children’s nutrition status in Uganda.

-Is the study design appropriate to address the stated objectives?

Yes, the study design is appropriate.

-Is the population clearly described and appropriate for the hypothesis being tested?

There is some ambiguity with respect to what the study population is. Is it “primary school going children living in recognised schistosomiasis high transmission hotspots in the West Nile sub-region district 166 of Pakwach“ (lines 164-165)? If so, then the extrapolation of prevalence to the whole district would be inaccurate (lines 22-293), as the study was not designed to estimate district prevalence. As well as this, if the population of interest was that of the hotspots, shouldn’t the two sites that were located away from the river have been excluded from the analyses?

It might be useful to define “hotspot” the first time this word is used (line 140).

-Is the sample size sufficient to ensure adequate power to address the hypothesis being tested?

Sample size, ignoring non-response rate, was 97 (not reported by the authors). Two sites had fewer participants than required (Pamitu 32, Alwi 62). No mention of this is made in the study.

The notation on the formula for sample size is confusing: ensure the alpha is subscripted (line 186) and remove “P<0.05” from line 188.

-Were correct statistical analysis used to support conclusions?

The statistical analyses used were not properly described. Please present a clear analysis plan, with sufficient description and justification of the methods used.

The authors are not consistent in their application of hierarchical modelling across tests.

Line 260: do the authors mean “overall” instead of “population prevalence”?

Line 261: do the authors mean “point” instead of “sample”?

Lines 261-264 need clarification. For instance, why do the authors estimate confidence intervals and credible intervals? etc.

Lines 268-269 also need clarification. For example, what is being achieved by this technique?

There is no information in the statistical methods section about how POC-CCA sensitivity and specificity were estimated. For example, using CCA as a gold standard; how were the confidence intervals estimated? Mention the limitations on this approach in the discussion.

There is insufficient information in the statistical methods section about the methods used in the nutritional section. For instance, the type of correlation used between BMI and MUAC needs to be specified; some results refer to F values for tests that aren’t specified, the inclusion of covariates in the model from Table S8 has not been justified, etc.

-Are there concerns about ethical or regulatory requirements being met?

Yes. Consent was obtained on day 2. This means that investigators did not have consent to process the POC-CCA samples upon collection on day 1. Hence, the results of the participants that did not give consent on day 2 may need to be removed from the study.

Other concerns:

Line 217: if the two technicians disagreed, how was this resolved?

Lines 211-223 and 227-233: if these lines are not relevant for this study, consider removing them.

Reviewer #2: Yes

**Results**

-Does the analysis presented match the analysis plan?

-Are the results clearly and completely presented?

-Are the figures (Tables, Images) of sufficient quality for clarity?

Reviewer #1: -Does the analysis presented match the analysis plan?

There was no clear analysis plan.

-Are the results clearly and completely presented?

Not always.

Prevalence:

• Why was prevalence based on POC-CCA results only? This needs justifying.

• No extrapolation to district can be made (lines292-294).

• Line 297: should it read “point” instead of “overall”?

• Lines 301-304: I am not certain that these results are needed.

Intensity of infection:

• Line 315: what does “confidence level 0.092” mean?

• Line 322: what does “confidence level 3.981” mean?

POC-CCA precision

• I suggest that this subheading be created.

• Line 359: expand on this: “The 6 KK positive cases that were negative by CAA did not greatly affect the specificity of the test”. How did the authors quantify this?

• Table S3: To avoid confusion between POC-CCA and CCA, I suggest that every time POC-CCA is referred to, the authors use the prefix “POC”.

• Table S3: there should be a difference between “Controls CCA (=0)” when trace is considered positive and negative, whereas in Table S3, in both cases, FN is 2 and TN is 21.

Nutrition status

• Line 362: Remove the word “impact” as this word implies a causal effect that was not assessed in this study.

• Line 367: does the F-test result cited refer to BMI or MUAC? How was this test done? If it was ANOVA, then the statement “the girls of the same age group showed higher levels (F = 1.7, P=8.12e-08) of both BMI and MUAC than the boys” would be incorrect.

• Line 373: What are “sites with control individuals”? Should these sites have been mentioned in the methods section?

• Line 385-386: this statement is misleading given the results presented in the next paragraph.

Other recommendations:

• Always report p-values with r values.

-Are the figures (Tables, Images) of sufficient quality for clarity?

On some graphs, the authors do not mention which estimate they are reporting, e.g. mean.

The word “phenotype” has not been consistently removed.

Reviewer #2: Yes

**Conclusions**

-Are the conclusions supported by the data presented?

-Are the limitations of analysis clearly described?

-Do the authors discuss how these data can be helpful to advance our understanding of the topic under study?

-Is public health relevance addressed?

Reviewer #1: -Are the conclusions supported by the data presented?

The conclusions are not fully supported by the data presented. 

• The authors do not fully explore the limitations of POC-CCA (they only focused on one of its limitations “misdiagnosis when calling trace results as positive”, but there are other relevant concerns); the authors do not explore the limitations of using CCA as a gold standard (particularly given that 6 KK + and CCA- cases were observed).

• The authors state that “… The impact of this is likely to be compounded by high levels of under nutrition and stunting especially among the boys”. However, this directly contradicts their finding that “we did not find any association between schistosomiasis infection and stunting” (line 483).

-Are the limitations of analysis clearly described?

The limitations of the analysis are insufficiently described. The authors only mention one limitation (lines 488-489).

Discussion

Line 458: this study was not designed to assess this statement.

Line 484: “we observed lower BMI levels in the schistosomiasis cases than controls” directly contradicts lines 369-371: “The average BMI for the S. mansoni infected cases was 17.09�2.04 and that of the uninfected controls was 15.5�2.39”.

Abstract

Lines 44-46: “Efforts to control schistosomiasis infection in Uganda have been greatly affected by … impact of disease on child growth”. We still don’t know what impact the disease has on child growth, hence we cannot be certain that this is having an impact on the efforts to control schistosomiasis.

Line 62: this study was not set up to assess “attribution”.

Line 64-65: the statement that “high levels of stunting might have an influence on morbidity” is misleading, particularly given the results of this study.

Title

“High levels of infection”; infection was primarily light (lines 322-323).

Reviewer #2: Yes

**Editorial and Data Presentation Modifications?**

Reviewer #1: (No Response)

Reviewer #2: In some elements of the text, the sentence structure requires revision to enhance readability.

**Summary and General Comments**

Reviewer #1: Please refer to my extensive comments in the sections above

Reviewer #2: The authors have addressed reviewer comments.

PLOS authors have the option to publish the peer review history of their article (what does this mean?). If published, this will include your full peer review and any attached files.

Reviewer #1: No

Reviewer #2: No
---

## [Editor Report · Decision Letter 2]

8 Jun 2022

Dear Dr. Matovu,

We are pleased to inform you that your manuscript 'High Prevalence of Schistosoma mansoni infection and stunting among school age children in communities along the Albert-Nile, Northern Uganda: a cross sectional study' has been provisionally accepted for publication in PLOS Neglected Tropical Diseases.

Best regards,

JOANNE P. WEBSTER

Associate Editor

Christine Budke

Deputy Editor

I will accept now, as much of value and interest here. However, I would still like to see further details re your definition of persistent schistosomiasis hotspot here (and indeed perhaps how this compares to the latest WHO definition given). Also ideally, whilst the title has improved, it would have been of value to clarify that no direct association between schistosome infection and stunting was observed here, although agreed both prevalent.

---

## [Editor Report · Acceptance letter]

21 Jul 2022

Dear Dr. Matovu,

We are delighted to inform you that your manuscript, "High Prevalence of Schistosoma mansoni infection and stunting among school age children in communities along the Albert-Nile, Northern Uganda: a cross sectional study," has been formally accepted for publication in PLOS Neglected Tropical Diseases.

Best regards,

Shaden Kamhawi

co-Editor-in-Chief

Paul Brindley

co-Editor-in-Chief
